# Dairy Consumption and the Colonic Mucosa-Associated Gut Microbiota in Humans—A Preliminary Investigation

**DOI:** 10.3390/nu17030567

**Published:** 2025-02-02

**Authors:** Ellie Chen, Nadim J. Ajami, Donna L. White, Yanhong Liu, Shawn Gurwara, Kristi Hoffman, David Y. Graham, Hashem B. El-Serag, Joseph F. Petrosino, Li Jiao

**Affiliations:** 1Department of Medicine, Baylor College of Medicine (BCM), Houston, TX 77030, USAdgraham@bcm.edu (D.Y.G.); hasheme@bcm.edu (H.B.E.-S.); 2The Alkek Center for Metagenomics and Microbiome Research, Department of Molecular Virology and Microbiology, Baylor College of Medicine (BCM), Houston, TX 77030, USA; 3Houston VA HSR&D Center for Innovations in Quality, Effectiveness and Safety, Michael E DeBakey Veterans Affairs Medical Center (MEDVAMC), Houston, TX 77030, USA; 4Texas Medical Center Digestive Disease Center, Houston, TX 77030, USA; 5Dan L Duncan Comprehensive Cancer Center, Baylor College of Medicine (BCM), Houston, TX 77030, USA; 6Section of Gastroenterology, Effectiveness and Safety, Michael E DeBakey Veterans Affairs Medical Center (MEDVAMC), Houston, TX 77030, USA

**Keywords:** diet, nutrition, microbiome, milk, cheese, lactose, *Faecalibacterium*, *Akkermansia*

## Abstract

Background: Dairy consumption has been associated with various health outcomes that may be mediated by changes in gut microbiota. Methods: This cross-sectional study investigated the association between the colonic mucosa-associated gut microbiota and the self-reported intake of total dairy, milk, cheese, and yogurt. A total of 97 colonic mucosal biopsies collected from 34 polyp-free individuals were analyzed. Dairy consumption in the past year was assessed using a food frequency questionnaire. The 16S rRNA gene V4 region was amplified and sequenced. Operational taxonomic unit (OTU) classification was performed using the UPARSE and SILVA databases. OTU diversity and relative abundance were compared between lower vs. higher dairy consumption groups. Multivariable negative binomial regression models for panel data were used to estimate the incidence rate ratio and 95% confidence interval for bacterial counts and dairy consumption. False discovery rate-adjusted *p* values (*q* value) < 0.05 indicated statistical significance. Results: Higher total dairy and milk consumption and lower cheese consumption were associated with higher alpha microbial diversity (adjusted *p* values < 0.05). Higher total dairy and milk consumption was also associated with higher relative abundance of *Faecalibacterium*. Higher milk consumption was associated with higher relative abundance of *Akkermansia*. Higher total dairy and cheese consumption was associated with lower relative abundance of *Bacteroides*. Conclusions: Dairy consumption may influence host health by modulating the structure and composition of the colonic adherent gut microbiota.

## 1. Introduction

Dairy products provide a rich source of nutrients, including protein, fats, carbohydrates, calcium, magnesium, selenium, phosphorous, potassium, zinc, and vitamins B2, B6, B12, D, and K [1,2]. However, the association between dairy consumption and the risk of various diseases, including cancer, remains inconclusive [3]. Fermented dairy products, in particular, have been shown to have many health benefits, such as improving bone health [4], building muscle mass, lowering blood pressure [5], and reducing the risk of cardiovascular disease and diabetes [6,7,8,9]. Additionally, moderate dairy consumption has been linked to improved cognitive function in later life [10]. On the other hand, higher consumption of dairy and milk has been associated with an increased risk of stroke, cardiovascular mortality, and all-cause mortality [11]. Given that dairy products are a heterogeneous food group comprising milk, cheese, yogurt, and butter, it is recommended to investigate the health effects of each individual food group separately.

Research has also examined the relationship between dairy consumption and molecular biomarkers in humans. A systematic review of intervention studies concluded that dairy and milk consumption did not exhibit pro-inflammatory effects in either healthy adults or adults with metabolic abnormalities, with long-term dairy supplementation showing a weak anti-inflammatory effect [12]. Another systematic review reported that higher dairy consumption, compared to lower or no consumption, was associated with reduced levels of pro-inflammatory biomarkers [13]. In postmenopausal women, higher dairy consumption, excluding butter, was associated with favorable profiles of lipids, insulin response, and inflammatory biomarkers [14]. Overall, biomarker-based studies do not indicate adverse effects of dairy consumption on inflammatory biomarkers [15]. Additionally, the previous findings suggested that specific dairy foods may uniquely influence these biomarkers [14]. Further research into the impact of dairy consumption on other biological functions could enhance our understanding of its potential health benefits or risks.

In humans, the gut microbiota is a dynamic physiological system that plays a critical role in various host functions, including nutrient metabolism, xenobiotic and drug metabolism, antimicrobial defense, immunomodulation, and maintaining the structural integrity of the gut mucosal barrier [16]. It is plausible that dairy consumption influences gut microbiota composition and diversity, particularly due to the probiotic effects of fermented dairy products such as yogurt and kefir [17,18]. A systematic review of eight studies concluded that milk, yogurt, and kefir may modulate the fecal microbiota composition in favor of the host [19]. In one randomized crossover study of 46 healthy overweight individuals, a high-dairy diet significantly altered fecal microbiota [20]. A study in men found no overall association between total dairy consumption and bacterial composition, though milk consumption was linked to higher relative abundance of *Ruminococcaceae* and *Bifidobacterium* [21]. A previous study found that consuming A2 milk increased *Bifidobacterium* spp. compared to consuming regular milk in humans [22]. These previous findings highlight the importance of investigating individual dairy foods in association with the gut microbiota. However, to date, the impact of dairy products on the mucosa-associated gut microbiota has not been examined.

Therefore, we investigated the relationship between the consumption of total dairy products, milk, cheese, and yogurt and the composition and structure of the colonic mucosa-associated gut microbiota in individuals with grossly normal colonoscopic findings. We hypothesized that dairy consumption affects the composition and structure of the gut microbiota.

## 2. Materials and Methods

### 2.1. Study Population

The research design, eligibility criteria, and data collection protocol were described previously [23]. This was a cross-sectional study of prospectively and consecutively enrolled veterans who had a scheduled colonoscopy between August 2013 and April 2017 at the Michael E. DeBakey Veterans Affairs Medical Center (MEDVAMC) in Houston, Texas. All participants provided written informed consent for study activities. The study protocol was approved by the Institutional Review Boards of Baylor College of Medicine (BCM) and MEDVAMC.

We excluded individuals with the following criteria: (1) a history of familial or hereditary colon diseases or inflammatory bowel disease; (2) invasive cancer, except for nonmelanoma skin cancer; (3) colorectal polyps within the past 3 years; (4) end-stage renal disease requiring dialysis; (5) severe mental disabilities; (6) surgery or hospitalization within the past year; (7) oral or systemic use of antibiotics within the past 3 months; (8) infection with hepatitis B virus, hepatitis C virus, HIV, or methicillin-resistant *Staphylococcus aureus*; or (9) bleeding disorders or use of anticoagulants. Individuals who had changed their dietary habits in the past 3 months were also excluded. Participants were instructed to discontinue routinely used medications 7 days prior to the procedure and antidiabetic medications 1 day prior.

Eligible participants were included if they had grossly normal-appearing colon mucosa on colonoscopy. During the study period, we enrolled 612 participants, 562 of whom completed the colonoscopy procedure; 172 were found to have a grossly normal colon, of whom 133 had colonic biopsies obtained for this study (1–6 biopsies from cecum to rectum). We sent mucosal samples from 69 participants for sequencing. Among them, 40 participants completed the Food Frequency Questionnaire (FFQ). Five were excluded because their energy intake was <800 or >5000 kcal per day. Thus, we included 99 mucosal samples from 35 participants who had sequencing data. Two samples from one individual were further excluded due to the poor sequencing quality. Therefore, we included a total of 97 colonic biopsies from 34 participants in the final analysis.

### 2.2. Data Collection

After obtaining informed consent, we collected participants’ lifestyle data using an interviewer-administered questionnaire. We also obtained the anthropocentric parameters from each participant. Dietary data were collected using a self-administered, 110-item validated Block FFQ -2005, which has been validated as a tool to assess diet–disease risk [24]. NutritionQuest calculated the average daily servings of nutrients and foods over the past year, adjusting for caloric intake using the density method [25]. We computed the healthy eating index (HEI) to assess diet quality [26].

### 2.3. Library Construction and 16S rRNA Gene Sequencing

The microbiome profiling was performed at the Alkek Center for Metagenomics and Microbiome Research (CMMR) at BCM as described previously [27,28]. The bacteria were examined by amplifying the V4 region of the highly conserved small-sub-unit ribosomal gene (16S rRNA) and sequencing on the MiSeq platform (Illumina, SD, CA, USA) using the 2 × 250 bp paired-end protocol. The V4 region was amplified by PCR using primers 515F (GTGCCAGCMGCCGCGGTAA) and 806R (GGACTACHVGGGTWTCTAAT). Each resulting amplicon set was barcoded with a unique 12 mer tag.

The read pairs were demultiplexed based on the unique molecular barcodes tagged during amplification, and reads were merged using USEARCH v7.0.1090. Chimeras or low-quality sequences were removed using USEARCH v7.0.1090 and UCHIME v4.2. The 16S rRNA gene sequences were clustered into the Operational Taxonomic Units (OTUs) at a similarity cutoff value of 97% using the UPARSE algorithm. The OTUs were mapped to the SILVA v128 database to determine taxonomies using UPARSE. The abundances were recovered by mapping the demultiplexed reads to the UPARSE OTUs. The downstream analyses were performed using the Agile Toolkit for Incisive Microbial Analyses (ATIMA) (Houston, TX, USA). The samples were rarefied to 1648 reads per sample, resulting in the loss of all negative controls and two mucosal samples from one individual. Therefore, we included a total of 97 colonic biopsies from 34 participants in the final analysis.

### 2.4. Data Analysis

To examine the distribution of demographic and lifestyle factors by higher vs. lower total dairy intake, Student’s *t*-test was used to analyze differences in continuous variables with a normal distribution. Fisher’s exact test was used for categorical variables because frequencies in at least one cell were lower than 5. The median value derived from 34 participants served as the cutoff to categorize intake as lower vs. higher. The ATIMA was used to identify trends in taxa abundance and diversity according to higher vs. lower total dairy, milk, cheese, and yogurt intake, respectively. We examined all taxa with relative abundance > 0.05%. The unclassified OTUs were included in the analyses. We used principal coordinates analysis plots (PCoA) with the Monte Carlo permutation test to estimate *p* values for beta-diversity and used the Weighted UniFrac as the distance matrix. The Mann–Whitney test was used to compare the relative abundance of bacterial taxonomic levels by the intake level. Multivariable negative binomial regression analysis was used to estimate the incidence rate ratio (IRR) and its 95% confidence interval (CI) of having the bacterial count in the higher intake group compared to the lower intake group, adjusting for age, race (non-Hispanic White, African American, and Hispanic), body mass index (BMI), smoking status (never, former, and current), alcohol use (never, former, and current), diabetes, hypertension, HEI score, and biopsy segment. We additionally included the individual nutrient variables listed in Table 1 in the model. To account for the repeated colonic biopsy sampling from the same individuals, we ran a negative binomial regression for panel data (random effect). The beta-diversity and the relative abundance of the bacterial taxa did not differ by colon segment (*p* values = 0.99). Nevertheless, we repeated the analysis using a single segment (sigmoid) in the sensitivity analysis.

All tests were two-sided. A *p* value < 0.05 in the general analysis or a false discovery rate (FDR)-adjusted *p*-value (*q* value) < 0.05 in the microbiota analysis indicated statistical significance, respectively. STATA 16.0 (STATA Corporation, College Station, TX, USA) was used for data analysis.

## 3. Results

The study participants had a mean age of 62.7 years (range: 55–71 years), with only one woman participant. The median calorie-adjusted daily intake included 0.57 servings of total dairy, 0.24 cups of milk, 0.27 milk-equivalent servings of cheese, and 0.002 cups of yogurt. Two participants consumed low-fat cheese, while the rest consumed regular cheese. The distribution of age, race, BMI, smoking status, and disease history was comparable between participants with higher vs. lower total dairy consumption. Those with higher total dairy intake had significantly greater consumption of calcium, vitamin D, riboflavin, cobalamin, saturated fat, and lactose (*p* values < 0.05). However, there were no differences in diet quality and vitamin B6 intake between the groups (Table 1). None of the participants reported a history of autoimmune disorders.

Higher total dairy and milk intake and lower cheese intake were associated with a higher alpha-diversity (*q* values < 0.05). Yogurt intake was not associated with alpha-diversity (Figure 1A–D).

The bacterial community composition, which was indicated by the beta-diversity, differed significantly based on higher vs. lower intake of total dairy, milk, cheese, and yogurt (yes vs. no) (Figure 2A–D).

Figure 3 and Appendix A show the differences in the relative abundance of major phyla by dairy consumption. Participants with higher consumption of total dairy products, cheese, and yogurt had a lower relative abundance of *Bacteroidetes* (*q* value < 0.05). In contrast, those with higher intake of total dairy products, milk, and cheese exhibited a higher relative abundance of *Verrucomicrobia* (*q* value < 0.05). Additionally, the relative abundance of *Fusobacteria* was greater among individuals with higher consumption of total dairy products and cheese compared to their counterparts (*q* values < 0.05).

Figure 4 and Appendix A show significant differences in the relative abundance of major bacterial genera (>0.05% relative abundance) by dairy consumption. Compared to their counterparts, individuals with higher consumption of total dairy and milk had a greater relative abundance of *Faecalibacterium* and *Haemophilus*. Those consuming more total dairy and cheese exhibited a higher relative abundance of *Bacteroides* and *Fusobacterium*. A higher intake of milk and cheese was associated with an increased relative abundance of *Akkermansia*. While *Parabacteroides* was more abundant among those consuming more milk, it was less abundant in those consuming more cheese. Additionally, individuals with higher cheese consumption had an increased relative abundance of *Escherichia* and a decreased relative abundance of *Subdoligranulum*. Finally, a higher intake of total dairy products was linked to a greater relative abundance of *Bifidobacterium* (all *q* values < 0.05) (Appendix A).

Multivariable negative binomial regression for panel data analysis confirmed a statistically significant positive association between *Faecalibacterium* and the intake of total dairy and milk, as well as between *Akkermansia* and milk intake. However, these associations were attenuated after adjusting for lactose intake in the models. The analysis also confirmed a statistically significant inverse association between *Bacteroides* and *Subdoligranulum* and cheese intake. Adjusting for known nutrients did not impact the incidence rate ratio (IRR) estimates. Additionally, total dairy intake was inversely associated with *Bacteroides* (Table 2).

Appendix A shows the genera that significantly differed in relative abundance based on the sigmoid sample. We observed the increased *Faecalibacterium* in those with higher total dairy and milk intake (*q* values < 0.05) and increased *Bifidobacterium* with higher total dairy intake and increased *Akkermansia* with higher milk intake.

## 4. Discussion

We characterized the gut microbiota using colonic mucosa specimens. Higher intake of dairy and milk, along with lower cheese intake, was associated with a greater bacterial alpha-diversity. Beta-diversity analyses showed differences in bacterial composition between higher vs. lower intake of all types of dairy products. Higher total dairy and milk consumption was associated with a greater relative abundance of *Faecalibacterium*, while milk consumption was associated with a higher relative abundance of *Akkermansia*. Conversely, higher total dairy and cheese consumption was associated with a lower relative abundance of *Bacteroides*. Higher cheese intake was also associated with a lower relative abundance of *Subdoligranulum*. Our study indicated that dairy and milk consumption promote the colonization of beneficial gut bacteria in the colonic mucosa in humans. The study findings were in agreement with those reported by a recent review [29].

Our findings showed that bacterial community composition differed between higher vs. lower dairy intake. Similarly, a prospective study of a Chinese population reported significant differences in fecal gut microbial community structure (beta-diversity) when comparing the highest with the lowest intake categories for total dairy, milk, and yogurt. Dairy consumption was also associated with a higher alpha-diversity [30]. However, a cross-sectional study of Australian men found no difference in the alpha-diversity of fecal microbiota across the dairy intake group. However, differences in beta-diversity were observed between higher vs. lower milk and yogurt consumption [21]. Overall, our study, which analyzed mucosal samples, aligns with other studies which showed that the consumption of milk, cheese, and yogurt would influence bacterial composition in feces.

Participants with higher dairy and milk intake had a greater relative abundance of *Faecalibacterium*, a Gram-positive anaerobic bacterium from the *Firmicutes* phylum and *Ruminococcaceae* family [31]. Interestingly, diverse strains of *Faecalibacterium* have been detected in dairy cow milk, highlighting its potential probiotic property [32]. *Faecalibacterium* produces butyrate and other short-chain fatty acids through the fermentation of dietary fiber and its anti-inflammatory property has been well recognized [31]. It is one of the most abundant human intestinal microbiota, with *Faecalibacterium prausnitzii* being the most abundant species [33]. Experimental research has shown that *F. prausnitzii* can ameliorate colorectal tumorigenesis [34]. In addition, *F. prausnitzii* has been found to be downregulated in individuals with coronary artery disease and anxiety [35]. The depletion of *Faecalibacterium* has been associated with an increased risk of inflammatory bowel disease [36]. Our observations may help explain the reduced risk of various diseases associated with higher consumption of non-fermented milk [37,38]. However, it is noted that one crossover study in overweight individuals reported a reduction in fecal *F. prausnitzii* with a high-dairy diet [20]. Future research should evaluate the impact of individual dairy foods on this key commensal bacterium and the respective physiological consequences.

Participants with higher milk intake had a greater relative abundance of *Verrucomicrobia* and its genus *Akkermansia* in their colonic mucosal samples. This finding is consistent with a previous crossover study in hyperinsulinemic individuals, which showed that higher dairy intake was associated with an increase in the butyrate-producing bacteria of the *Firmicutes* and the *Verrucomicrobia* phyla compared to adequate dairy intake, with both negatively correlated with insulin resistance [39]. In addition, a study in infants found that mare’s milk consumption led to a higher relative abundance of *Akkermansia* in fecal samples [40]. *Akkermansia* is a mucin-degrading, Gram-negative, obligate anaerobic bacterium [41] that has been associated with a healthier metabolic profile through improved glucose and lipid metabolism [42]. It has also been associated with enhanced gut barrier function, reduced age-related mucosal thinning, and decreased inflammation, contributing to improved colon health [43,44]. However, in fiber-deprived conditions, *Akkermansia* has been shown to promote colonic inflammation [45]. Further study is needed to explore optimal strategies for enhancing the health-promoting roles of *Akkermansia* across diverse populations.

*Faecalibacterium* and *Akkermansia* are two abundant commensal bacteria in the human gut that play crucial roles in immune-related diseases [46]. Our study showed that the abundances of these two bacteria are influenced by dairy or milk products. It is noted that adjusting for lactose attenuated the association between total dairy and milk intake and the relative abundance of *Akkermansia* and *Faecalibacterium*, suggesting that nutrients in dairy products may possess prebiotic properties.

Previous research has highlighted the potential health benefits of yogurt consumption [47], including the inverse association with the risk of cardiovascular diseases and type 2 diabetes [48]. In a study of 130 healthy individuals, natural yogurt consumption was associated with higher fecal levels of *Akkermansia* compared to non-consumers, while sweetened yogurt intake was associated with lower levels of *Bacteroides* [49]. However, several other human studies did not observe changes in *Akkermansia* abundance associated with yogurt consumption [50,51,52]. Due to the low yogurt consumption in our study population, we could not draw a meaningful conclusion. *Bifidobacterium*, a common yogurt starter known for its probiotic function [53], had a low relative abundance in our study samples, as did *Lactobacillus* and *Streptococcus*, two other widely used yogurt culture starters [54]. These bacteria have been shown to be transiently increased in the gut microbiome in yogurt consumers [50,55].

Higher cheese intake was associated with a lower relative abundance of *Subdoligranulum*, a spore-forming, butyrate-producing bacterium from the *Firmicutes* phylum and *Ruminococcaceae* family [56]. One study found that *Subdoligranulum* was more common in the healthy mucosa compared to colon polyps [57], while other research reported its enrichment in patients with gastrointestinal neoplasms [58]. Additionally, *Subdoligranulum* has been linked to chronic inflammation and poor metabolic control in children with irritable bowel syndrome [59].

Higher total dairy and cheese intake were also associated with a lower relative abundance of *Bacteroides*, the most abundant genus in our study population. *Bacteroides* produce toxins that can induce tumorigenesis and have been linked to colon cancer development [60,61]. Conversely, lower levels of *Bacteroides* in the gut have been associated with inflammatory bowel disease [62]. Taken together, these findings provide conflicting evidence regarding whether *Subdoligranulum* and *Bacteroides* are beneficial or detrimental to the host. Similarly, epidemiologic studies on the association between cheese intake and colorectal cancer (CRC) have yielded inconsistent results. Some research has reported an inverse association between cheese intake and CRC risk [63], while others have found no significant association [37,64]. A deeper understanding of the functional roles of *Subdoligranulum* and *Bacteroides* is essential for interpreting these findings and their implications for health.

The differing impact of milk and cheese consumption on gut bacterial composition may be explained by differences in their production and nutrient content. Cheese production involves fermentation, during which enzymes or bacteria break down lactose and produce lactic acid [65]. Milk contains a higher amount of lactose [66], which can act as a prebiotic that promotes the growth of a cancer-protective colonic microbiota [67,68]. In contrast, hard cheese contains very little lactose. Therefore, turning milk into cheese may alter its nutrient composition, potentially modifying its prebiotic or probiotic effects. However, in our study samples, the relative abundances of lactic-acid bacteria, including *Streptococcus*, *Veillonella*, and *Bifidobacterium*, were all below 0.5%.

Our study had multiple strengths. First, we examined the association between colonic adherent bacteria and dairy consumption. Mucosa-associated bacteria are thought to have a stronger connection to the host’s immune function compared to luminal bacteria [69]. Second, we implemented stringent eligibility criteria aiming at minimizing potential confounding factors such as antibiotics use and other diseases. Third, we accounted for multiple covariates when analyzing the association between bacteria and dairy consumption. Lastly, data collection and 16S rRNA sequencing were conducted and analyzed consistently, reducing the potential for systemic bias.

This study also had several limitations. First, we could not infer causality in the association between dairy consumption and alterations in gut microbiota given our cross-sectional design. Second, our participants were mostly veteran men over the age of 50, limiting the generalizability of the findings to women and other populations. For example, our participants reported lower average dairy consumption compared to older men in the National Health and Nutrition Examination Survey (NHANES), with average daily intake of total dairy, milk, and cheese being 1.80, 1.00, and 1.70 servings, respectively [70], compared to 0.56, 0.22, and 0.26 servings in our study. Future studies are warranted that can investigate these associations in populations with higher dairy consumption. According to MyPlate by the United States Department of Agriculture, adult men are recommended to consume three cups of dairy products daily. Third, potential information bias in the FFQ data cannot be ruled out. Fourth, although all participants had normal colonoscopy results, we were unable to assess other functional disorders that might have influenced the colonic gut microbiota [71]. Lastly, our final sample size was smaller than feces-based studies due to the use of mucosal samples. Comparisons between our mucosa-based findings and previous feces-based findings may not be valid. Our study findings should be confirmed in other large studies using mucosal samples.

## 5. Conclusions

Variations in dairy consumption were associated with distinct community composition and structure of gut microbiota attached to the colon. The potential beneficial effects of dairy and milk consumption on the gut microbiota community composition and structure aligned with the observed inverse association between dairy intake and the risk of various diseases. It is noted that milk and cheese intake were associated with the gut microbiota differentially. Further research is needed to explore the roles of less abundant bacteria in dairy consumption and their impact on health outcomes. Genomic, metagenomic, and metabolomic approaches are needed to identify bacterial species and functions influenced by dairy consumption and to identify the populations that could benefit from dairy consumption through personalized nutritional intervention [72]. In addition, targeted fecal microbiota transplantation using the optimized strains could be a promising preventive and treatment tool for various diseases [73].

## Figures and Tables

**Figure 1 nutrients-17-00567-f001:**
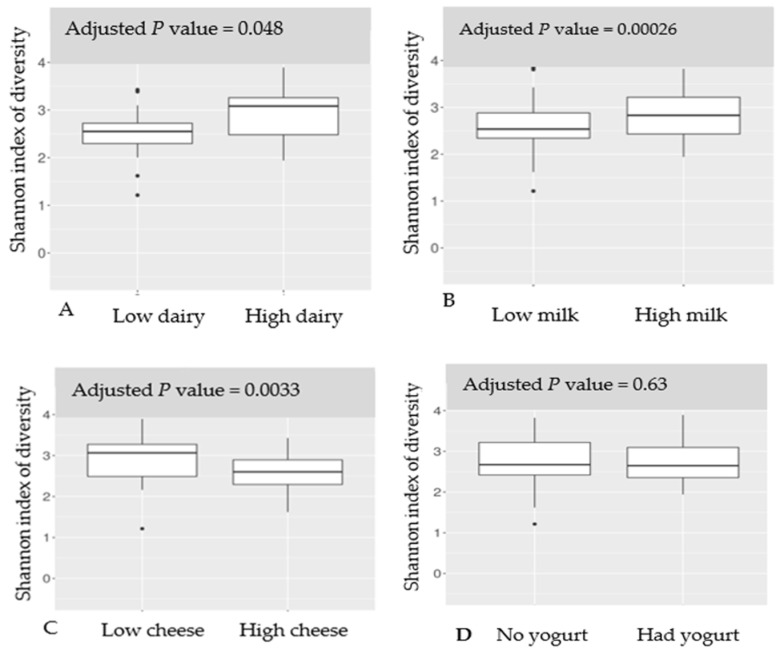
The alpha-diversity of the gut bacteria by higher vs. lower intake of total dairy (**A**), milk (**B**), cheese (**C**), and yogurt (no vs. yes) (**D**). Comparisons of bacterial Shannon index (*Y*-axis) were performed using the Mann–Whitney test between higher versus lower intake of each dairy product (*X*-axis). Adjusted *p* value indicates the significance of the difference.

**Figure 2 nutrients-17-00567-f002:**
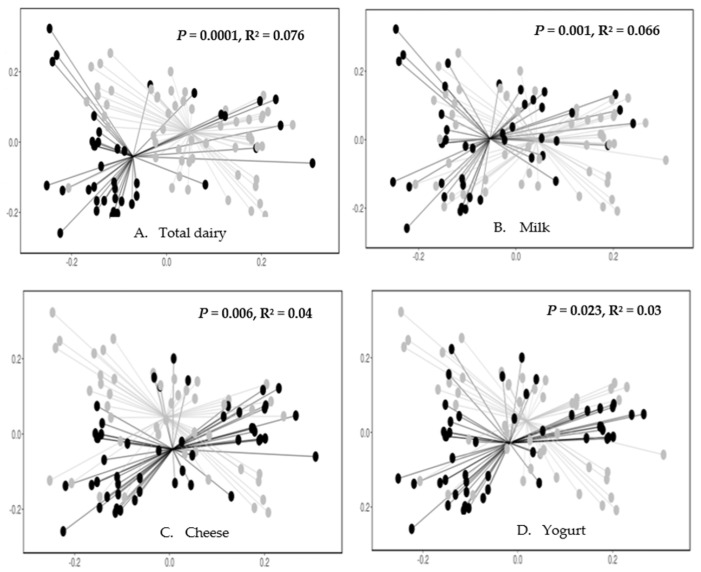
The beta-diversity of the gut bacteria by higher vs. lower intake of total dairy (**A**), milk (**B**), cheese (**C**), and yogurt (no vs. yes) (**D**). The principal coordinate plot used the Weighted UniFrac as the distance matrix. The Monte Carlo permutation test was used to estimate *p* values. Each symbol represents a sample. The lower intake group (black) was separated from the higher intake group (gray). The proportion of variance explained by the first two principal coordinates was denoted in the corresponding axis label. For all the panels, the *X*-axis is PC1 (principal component 1, explaining 33.2% of the variation); the *Y*-axis is PC2 (principal component 2, explaining 25.9% of the variation).

**Figure 3 nutrients-17-00567-f003:**
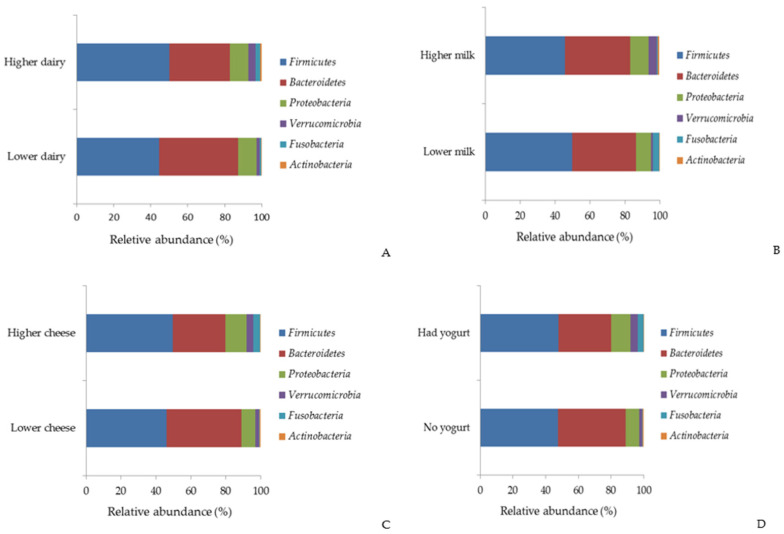
Relative abundance (%) of the major bacterial phyla by total dairy (**A**), milk (**B**), cheese (**C**), and yogurt (**D**).

**Figure 4 nutrients-17-00567-f004:**
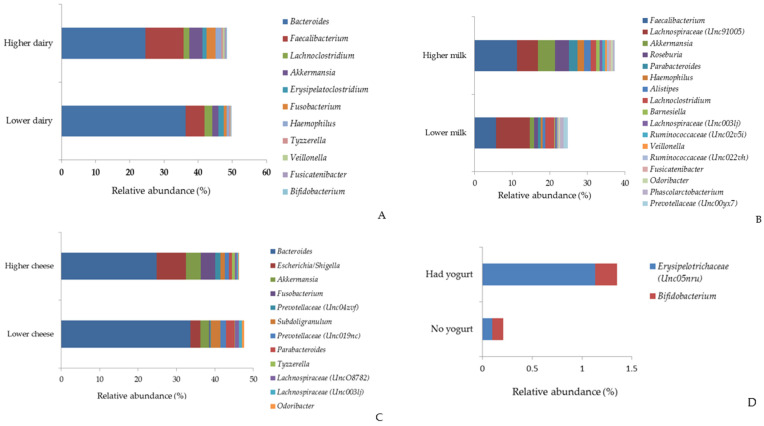
Relative abundance (%) of major bacterial genera by total dairy (**A**), milk (**B**), cheese (**C**), and yogurt (**D**). (Only the major genera (>0.5% relative abundance) with statistically significant differences (*q* values < 0.05) are shown in the figure).

**Table 1 nutrients-17-00567-t001:** Characteristics of the study participants by total dairy intake.

Characteristics Mean ± Standard Deviation or n (%)	Lower Dairy Intake n = 17	Higher Dairy Intake n = 17	*p* Value
Age (years)	62.7 ± 6.0	61.3 ± 5.1	0.50
Male, n (%)	16 (94%)	17 (100%)	0.31
Racial group			
White/Caucasian, n (%)	13 (76.5)	11 (64.7)	0.86
African American, n (%)	2 (11.8)	4 (23.5)	
Hispanic, n (%)	2 (11.8)	2 (11.8)	
Body mass index (kg/m^2^)	33.2 ±6.7	34.6 ±6.3	0.54
Smoking Status			
Never, n (%)	6 (35.3)	7 (41.2)	1.00
Former, n (%)	7 (41.2)	7 (41.2)	
Current, n (%)	4 (23.5)	3 (17.6)	
Alcohol Status			0.22
Never drinker, n (%)	4 (23.5)	5 (29.4)	
Former drinker, n (%)	3 (17.6)	7 (41.2)	
Current drinker, n (%)	10 (58.9)	5 (29.4)	0.46
Hypertension, yes (%)	13 (76.4)	12 (70.6)	0.71
Type 2 diabetes, yes (%)	9 (52.7)	8 (47.1)	1.00
Total HEI score	60.6 ± 9.66	61.2 ± 8.50	0.84
Calcium (mg/day/1000 kcal)	316 ± 69.6	460 ± 90.7	< 0.0001
Vitamin D (IU/day/1000 kcal)	44.5 ± 24.9	96.9 ± 68.4	0.006
Riboflavin (mg/day/1000 kcal)	0.92 ± 0.26	1.24 ± 0.29	0.002
Vitamin B6 (mg/day/1000 kcal)	0.87 ± 0.28	0.96 ± 0.24	0.31
Cobalamin (µg/day/1000 kcal)	2.00 ± 0.77	2.95 ± 0.66	0.0005
Saturated fat (g/day/1000 kcal)	11.3 ± 1.61	14.9 ± 3.23	0.0002
Lactose (g/day/1000 kcal)	2.52 ± 1.60	7.24 ± 5.35	0.001
Milk intake (cup/Day)	0.14 ± 0.11	0.49 ± 0.42	0.002
Cheese (serving/Day)	0.22 ± 0.03	0.35 ± 0.05	0.03
Yogurt intake (cup/Day)	0.003 ± 0.001	0.049± 0.08	0.031

HEI, healthy eating index; *p* value for Student’s *t* test for continuous variables or Fisher’s exact test for categorical variables.

**Table 2 nutrients-17-00567-t002:** The incidence rate ratio and 95% confidence interval of having bacteria count in mucosal samples by higher vs. lower intake of dairy products.

	Prevalence, %		Median Count		IRR (95% CI) ^1^	IRR (95% CI) ^2^	IRR (95% CI) ^3^
	Higher	Lower	Higher	Lower			
Total dairy	n = 40	n = 57	n = 40	n = 57			
*Akkermansia*	64.9	50.0	17.0	0.5	7.37 (1.79–30.3)	8.09 (2.15–30.3)	1.69 (0.31–9.20)
*Faecalibacterium*	100	85.0	159	45	4.70 (2.52–8.75)	4.88 (2.56–9.30)	5.00 (2.10–11.9)
*Bifidobacterium*	10.0	26.3	0	0	7.81 (1.17–52.2)	7.98 (1.32–48.0)	5.32 (0.76–37.2)
*Bacteroides*	100	100	595	404	0.61 (0.39–0.96)	0.61 (0.39–0.96)	0.43 (0.24–0.79)
Milk intake	n = 54	n = 43	n = 54	n = 43			
*Akkermansia*	81.5	30.2	23.5	0	28.6 (7.72–106)	25.4 (5.83–110)	7.18 (1.26–40.9)
*Faecalibacterium*	100	86.0	156	66	4.70 (2.24–9.86)	4.16 (1.97–8.78)	2.89 (1.21–6.90)
Cheese intake	n = 46	n = 51	n = 46	n = 51			
*Bacteroides*	100	100	338	554	0.52 (0.31–0.85)	0.50 (0.30–0.82)	-
*Subdoligranulum*	89.1	100	7	38	0.46 (0.29–0.74)	0.54 (0.34–0.87)	-

CI: Confidence interval. IRR: incidence rate ratio. ^1^ Negative binomial regression model for panel data used the bacterial count as the dependent variable. The food intake was modeled as higher vs. lower. The lower intake was the reference group. The model was adjusted for age, race (non-Hispanic White, African American and Hispanic), body mass index, smoking status (never, former, and current), alcohol use (never, former, and current), diabetes, hypertension, HEI score, and biopsy segment. The incidence rate of *Faecalibacterium* in those who had a higher milk intake was 4.16 times that of those who had a lower milk intake. ^2^ For milk or dairy intake, the model was further adjusted for cobalamin. For cheese intake, adjustment for nutrients did not further attenuate the IRR estimates. ^3^ For milk or dairy intake, the model was further adjusted for lactose based on model 1.

## Data Availability

Data available on request due to local policy on privacy.

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
