# Peer review of "Dairy Consumption and the Colonic Mucosa-Associated Gut Microbiota in Humans—A Preliminary Investigation"

_nutrients, 2025, doi:10.3390/nu17030567_

Round 1

Reviewer 1 Report

Comments and Suggestions for Authors

Recommendations:

1. How does probiotic supplementation perform in comparison with fecal microbiota transplantation. see this:https://doi.org/10.3390/jcm13216578 

2. Also does supplementation with probiotics could alleviate patterns that lead to different disease? https://doi.org/10.3390/diagnostics14090861 also see this.

3. No discussion about the impact on Short chain fatty acid producing flora? 

4. Student t-test is used only for normally distributed data, also Fisher's exact test is used only for expected frequency lower than 5, other-wise pearson chi-square should be used instead.

5. Normal endoscopy does not exclude functional disease, how the authors respond to this.

6. Normal endoscopy does not exclude factors that alter microbiome like infection, antibiotic use, diet,. 

Author Response

Comment 1: How does probiotic supplementation perform in comparison with fecal microbiota transplantation. see this:https://doi.org/10.3390/jcm13216578 

Response 1: This is a very interesting and valid scientific question. The provided literature underscores the promising of using FMT in colorectal cancer treatment. Our line of research underscores the importance of diet in shaping the colonic microbiota in humans. Both approaches could be used to modulate immune response and tissue microenvironment in the gut.

The present research is not designed to answer the question the reviewer proposed. Nevertheless, we acknowledged the FMT as the treatment tool for modulating microbiota and potentially the immune response in humans. This literature was cited as the last sentence of this manuscript (reference 70).

Thank you.

Comment 2: Also does supplementation with probiotics could alleviate patterns that lead to different disease? https://doi.org/10.3390/diagnostics14090861 also see this.

Response 2: Thank you for the intriguing question. The present research is not designed to answer the question the reviewer proposed. However, we acknowledged the importance of the provided research on FMT as we addressed to the first point. One literature was cited in as the last sentence of this manuscript.

       Comment 3: No discussion about the impact on Short chain fatty acid producing flora? 

Response 3: Thank you for the comments. We stated that “Faecalibacterium is a butyrate-producing” in the original version. In the revision, we revised the statement as “Faecalibacterium produces butyrate and other short-chain fatty acids through the fermentation of dietary fiber and its anti-inflammatory property has been well recognized [30]”. Please see line 5 of the third paragraph in Discussion on Page 11.

Comment 4: Student t-test is used only for normally distributed data, also Fisher's exact test is used only for expected frequency lower than 5, other-wise pearson chi-square should be used instead.

Response 4: Thank you for the comments. The paragraph has changed to “To examine the distribution of demographic and lifestyle factors by higher or lower total dairy intake, the Student’s t-test was used to analyzed differences in continuous variable with a normal distribution. Fisher’s exact test was used for categorical variables because the frequency in at least one cells was lower than 5. A median value derived from 34 participants served as the cutoff to categorize intake as lower or higher.” Please see the first paragraph in Section 2.4.

Comment 5: Normal endoscopy does not exclude functional disease, how the authors respond to this.

Response 5: Thank you for the comment. We agree that the functional gastrointestinal disorders may relate to gut microbiota under a normal endoscopy. We included this insight as one of the limitations in the manuscript. It states “Fourth, Although all participants had normal colonoscopy results, we were unable to assess unknown functional disorders that might have influenced the colonic gut microbiota [69]. “

Comment 6: Normal endoscopy does not exclude factors that alter microbiome like infection, antibiotic use, diet,. 

Response 6: Thank you for the comment. In order to investigate the association between diet and gut microbiota, we did apply multiple exclusion criteria in our research that have been previously reported (reference 22). Nevertheless, we added the second paragraph to 2.1 in responding to the reviewer’s concern.  It states “We excluded individuals with the following criteria: 1) a history of familial or hereditary colon diseases or inflammatory bowel disease; 2) invasive cancer, except for nonmelanoma skin cancer; 3) colorectal polyps within the past 3 years; 4) end-stage renal disease requiring dialysis; 5) severe mental disabilities; 6) surgery or hospitalization within the past year; 7) oral or systemic use of antibiotics within the past 3 months; 8) infection with hepatitis B virus, hepatitis C virus, and HIV, or methicillin-resistant Staphylococcus aureus; or 9) bleeding disorders or use of anticoagulants. Individuals who had changed their dietary habits in the past 3 months were also excluded. Participants were instructed to discontinue routinely used medications 7 days prior to the procedure and antidiabetic medications 1 day prior.”

Reviewer 2 Report

Comments and Suggestions for Authors

The manuscript investigates the association between dairy consumption and the composition of colonic mucosa-associated gut microbiota in humans. The study is novel, focusing on mucosa-associated microbiota rather than fecal microbiota. It is well organized, with clear objectives and appropriate use of statistical tools to analyze the data. The findings are relevant to understanding the relationship between diet and gut microbiota, a topic of growing interest in nutrition and health research.

Major comments:

Please provide the primer sequences in PCR of the 16S library construction.

The bacterial names need to be italic in the main manuscript.

Consider using more descriptive axis labels in figures, e.g., "Shannon Index of Diversity" instead of "Y-axis" or "Alpha Diversity".

Clarify the choice of statistical tests in cases where multiple options were available.

The format of the P values (q values) should be consistent across figures and tables. Make sure if the letters need to be italic or uppercase.

Table 2 and Table 3 are recommended to be replaced with stacked bar plots, which are more informative and intuitive.

Comments on the Quality of English Language

-

Author Response

Comment 1: Please provide the primer sequences in PCR of the 16S library construction.

Response 1: The primer sequence has been added. “The V4 region was amplified by PCR using primers 515F (GTGCCAGCMGCCGCGGTAA) and 806R (GGACTACHVGGGTWTCTAAT).” Please see the end of the first paragraph in Section 2.3.

Comment 2: The bacterial names need to be italic in the main manuscript.

Response 2: The changes have been made to all bacteria name in the manuscript. Thank you for helping improve the consistency.

Comment 3: Consider using more descriptive axis labels in figures, e.g., "Shannon Index of Diversity" instead of "Y-axis" or "Alpha Diversity".

Response 3: The change has been made to Figure 1 as suggested in the revision. Thank you.

Comment 4: Clarify the choice of statistical tests in cases where multiple options were available.

Response 4: Thank you. We have clarified the use of student’s t test and Fisher’s exact test in the beginning of section 2.4. This is also the concern of the reviewer 2.  

Comment 5: The format of the P values (q values) should be consistent across figures and tables. Make sure if the letters need to be italic or uppercase.

Response 5: The P value is italic and uppercase and q value is now italic. Thank you for helping us in enhancing consistency.

Comment 6: Table 2 and Table 3 are recommended to be replaced with stacked bar plots, which are more informative and intuitive.

Response 6: Previous Table 2 and Table 3 have been updated to Figure 3 and Figure 4, respectively. The previous Table 2 and Table 3 are now included as supplemental table 1 and 2 in the revision. Thank you.

Reviewer 3 Report

Comments and Suggestions for Authors

The paper is very interesting but I have some suggestions:

- At page 3, in data analysis. should be also inserted autoimmune disorders

- At page 4, the Authors, in Results, should better explain because in this study is present only 1 woman

-At page 6, the Authors, should better describe the growth of  protective and dangerous bacteria related to ingestion of dairy products

Author Response

Comment 1: The paper is very interesting but I have some suggestions.

Response 1: Thank you for positive comment!

Comment 2: At page 3, in data analysis. should be also inserted autoimmune disorders

Response 2: Thank you for the question! We did include autoimmune disorders in our survey. However, none of the study participants reported having autoimmune disorders. We added this finding in the Table 1 portion. “None of the participants reported a history of autoimmune disorders.” Please see the last sentence of the first paragraph under Results (Page 4).

Comment 3: At page 4, the Authors, in Results, should better explain because in this study is present only 1 woman.

Response 3: Thank you for the query. In this veteran-based study, we indeed included one woman in this study. We added this sentence in the limitation portion. “Second, our participants were mostly veteran men over the age of 50, limiting the generalizability of the findings to women and other populations..” Please see the 4th paragraph on Page 13.

Comment 4: -At page 6, the Authors, should better describe the growth of protective and dangerous bacteria related to ingestion of dairy products.

Response 4: Thank you for the comment. “Our study indicated that dairy and milk consumption promotes the colonization of beneficial gut bacteria in the colonic mucosa” (the last sentence of the first paragraph of Discussion (page 11). However, the harmful bacteria are not well unknown. Therefore, we did not comment on that.  We have a comment at the end of the 4th paragraph on page 15. “A deeper understanding of the functional roles of Subdoligranulum and Bacteroides is essential for interpreting these findings and their implications for health.  “

Round 2

Reviewer 1 Report

Comments and Suggestions for Authors

Congratulations!

Author Response

Comment 1: The authors answered the questions from the Reviewers in a satisfactory way. Only a bit minor questions should be addressed in the manuscript prior to its final acceptance.

Response 1: Thank you.

Comment 2: Abstract “adjusted P values” only should be in italics “P”

Response 2: Thank you for catching this issue. We fixed the italics.  

Comment 3: Figures 1 A to D should be aligned, and not bold letters should be used in figures.

Response 3: We have realigned the Figure 1, A to D. Please let us know whether it is fine.

Comment 4: Figure 4D: genus names should be in italics

Response 4: We apologized for missing this format. The updated figure has been included in the main text and Figure document. Thank you for catching it.

Comment 5: Discussion: « Our study indicated that dairy and milk con-sumption promotes the colonization of beneficial gut bacteria in the colonic mucosa in human”..and this was in agreement with those reported by a recent review (Mondragon et al. Substitutive Effects of Milk vs. Vegetable Milk on the Human Gut Microbiota and Implications for Human Health).

Response 5: This statement and the reference (reference 29) have been added to the second revision. Thank you for the comments. Page 11, line 3.

Comment 6: Regarding milks, it would be interesting to cite that A2 milk increases Bifidobacterium counts higher than regular cow milk (Fernadez-Rico et al. A2 Milk: New Perspectives for Food Technology and Human Health).

Response 6: We added this statement to the 3rd paragraph on page 2 in Introduction. “A previous study found that consuming A2 milk increased Bifidobacterium spp. compared to consuming regular milk in humans [45]. “ The suggested reference is added (reference 22). Thank you.
